# All-optical control and super-resolution imaging of quantum emitters in layered materials

Mehran Kianinia[1], Carlo Bradac[1], Bernd Sontheimer [2], Fan Wang [1], Toan Trong Tran[1]
Minh Nguyen[1], Sejeong Kim[1], Zai-Quan Xu[1], Dayong Jin [1], Andreas W. Schell[3], Charlene J. Lobo[1],
Igor Aharonovich[1] & Milos Toth [1]

Layered van der Waals materials are emerging as compelling two-dimensional platforms for nanophotonics, polaritonics, valleytronics and spintronics, and have the potential to transform applications in sensing, imaging and quantum information processing. Among these, hexagonal boron nitride (hBN) is known to host ultra-bright, room-temperature quantum emitters, whose nature is yet to be fully understood. Here we present a set of measurements that give unique insight into the photophysical properties and level structure of hBN quantum emitters. Specifically, we report the existence of a class of hBN quantum emitters with a fast-decaying intermediate and a long-lived metastable state accessible from the first excited electronic state. Furthermore, by means of a two-laser repumping scheme, we show an enhanced photoluminescence and emission intensity, which can be utilized to realize a new modality of far-field super-resolution imaging. Our findings expand current understanding of quantum emitters in hBN and show new potential ways of harnessing their nonlinear optical properties in sub-diffraction nanoscopy.

[1] School of Mathematical and Physical Sciences, University of Technology Sydney, Ultimo, NSW 2007, Australia. [2] Institut für Physik, Humboldt-Universität zu Berlin, 12489 Berlin, Germany. [3] Department of Electronic Science and Engineering, Kyoto University, 615-8510 Kyoto, Japan. Mehran Kianinia and Carlo Bradac contributed equally to this work. Correspondence and requests for materials should be addressed to C.B. (email: carlo.bradac@uts.edu.au) or to I.A. (email: igor.aharonovich@uts.edu.au) or to M.T. (email: milos.toth@uts.edu.au)

The exploration of nanophotonic phenomena in two-dimensional (2D) systems using materials such as transition metal dichalcogenides, phosphorene and hexagonal boron nitride (hBN) has gained considerable momentum in recent years[1–11]. Localized nanoscale effects, including radiative decay of interlayer excitons and emission of anti-bunched photons from deep trap point defects, are particularly interesting and important[11–13]. The latter, for instance, is key to the practical deployment of scalable, on-chip quantum photonic devices[14,15]. In this context, 2D-layered hBN has shown great promise owing to its ability to host fully polarized, ultra-bright and narrow-linewidth colour centres, which act as photostable quantum emitters at and beyond room temperature[9–11,16–18]. Their nature is still under dispute, as is a convincing explanation for the large distribution of observed zero-phonon line (ZPL) energies (ranging from the ultraviolet to the near-infrared)[17,19] and for the photodynamic properties of the emitters.

In this work, we carry out a series of systematic experiments designed to shed light onto the level structure and photo-dynamics of quantum emitters in hBN. We perform room temperature, off-resonant excitation at different wavelengths, as well as second-order autocorrelation and saturation measurements. By employing two-laser excitation, we identify an entire class of hBN emitters that possess unique photo-dynamics, allowing the emitters to be reverted from the intermediate state to the excited state(s) by means of optical repumping. This leads to a strong nonlinear enhancement of photoemission and a reduction of the excitation power that is needed to saturate the emission intensity. We exploit this highly nonlinear behaviour to propose and demonstrate a new scheme for super-resolution imaging.

## Results

**Quantum emitters in hBN.** A schematic of the hBN atomic lattice is shown in Fig. 1a, with blue and pink corresponding to nitrogen and boron atoms, respectively. The atomic structure of the quantum emitter(s) is still a matter of debate and a number of vacancy-related defects have been proposed in literature[11,20,21]. We started by surveying several emitters using a conventional confocal, optical microscope and a Hanbury–Brown and Twiss (HBT) interferometer. For reference, Fig. 1b shows the room temperature photoluminescence (PL) spectrum of one such emitter when excited with a 675 nm wavelength laser. The emitter has a ZPL at 778 nm and a negligible phonon sideband. Figure 1c displays the second-order correlation function, $g^{(2)}(\tau)$, which indicates that the emission is predominantly from a single defect: $g^{(2)}(\tau = 0) \approx 0.25$, well below 0.5 at zero delay time (the correlation data are not background-corrected). Figure 1d shows the saturation behaviour of the emitter, excited with a 708 nm laser. The data have been fitted to the equation $I = I_{\infty} \times P/(P + P_{sat})$; accordingly, the 50% value of the saturated emission intensity occurs at 14 mW (cf. Methods). Polarization measurements (Fig. 1d, inset) reveal that the photoemission of the centre is fully polarized, as expected from a dipole located in-plane within the layered host crystal and consistent with the proposed chemical structure of the defect(s).

After confirming the quantum nature of the surveyed emitters, we introduced a second laser to look for nonlinearities in the PL. Figure 1e displays the emission intensity for the reference emitter upon excitation with the following: a 675 nm laser (purple trace), a 532 nm laser (green trace) and the simultaneous pair of 675 nm + 532 nm lasers (red trace). The emitter was excited with 10 μW

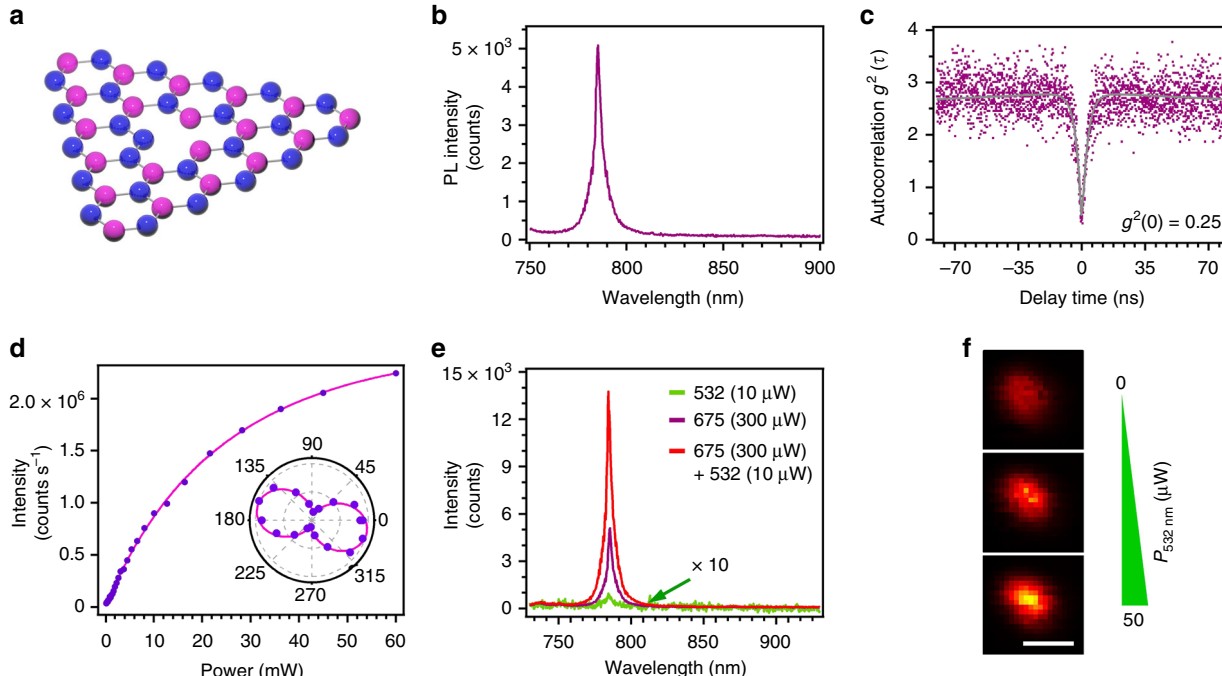

**Fig. 1** Single photon emission from hBN. **a** Two-dimensional hBN hosting a fluorescent defect. **b** Photoluminescence spectrum of the single defect in hBN under 675 nm excitation. **c** Second-order autocorrelation measurement $g^{(2)}(\tau)$ of the defect with a dip at $g^{(2)}(0) \approx 0.25$, indicating single photon emission. **d** Saturation curve of the emitter under excitation with a 708 nm laser. The solid line is the fit to $I = I_{\infty} \times P/(P + P_{sat})$, according to which the 50% of the saturated emission occurs at 14 mW. Inset: emission polarization curve of the emitter. The emission dipole is in the plane of the layered host crystal. (Note: the 708 nm laser was used in lieu of the 675 nm one for measurements requiring high power). **e** Photoluminescence spectra of the single defect under three excitation conditions: low-power (10 μW) 532 nm laser (green trace), high-power 675 nm laser (purple trace) and coincident excitation with both lasers (red trace). It is noteworthy that the spectrum under excitation with the 532 nm laser (green trace) has been multiplied tenfold for display purposes. **f** Nonlinear increase of the emitter brightness upon a linear increase of the power of the 532 nm excitation laser coincident with the 675 nm excitation laser. Scale bar is 500 nm

of power for the 532 nm laser and 300 μW for the 675 nm laser. The 532 nm excitation yielded a negligible fluorescence intensity —it is noteworthy that the corresponding PL spectrum in Fig. 1e (green trace) is multiplied tenfold for clarity. Interestingly, comparing the excitation of the emitter with the 675 nm laser (purple trace) and the co-excitation with the laser pair, i.e., 675 nm (300 μW) plus 532 nm (10 μW) (red trace), reveals a highly nonlinear behaviour (Fig. 1e). Specifically, upon co-excitation, the emission intensity increases by more than twofold, which is far greater than the 3.3% increase in total excitation power (from 300 μW to 310 μW). This behaviour, highlighted in Fig. 1f, is attributed to repumping of the emitter by the 532 nm laser, which repopulates the excited bright state from the intermediate state (see below). The emitter does not bleach for laser excitation powers as high as 60 mW, which is used to obtain the saturation curve in Fig. 1d. The same nonlinear behaviour was observed on other—yet not all—emitters (cf. Supplementary Note 1 and Supplementary Figures 1–2) with different ZPL energies < 1.77 eV (i.e., emission wavelengths > 700 nm).

**Photo-physics and nonlinear behaviour of emitters in hBN**. Figure 2a shows long-time (up to milliseconds) second-order autocorrelation measurements, $g^{(2)}(\tau)$, recorded for the reference quantum emitter introduced in Fig. 1. The different data sets are collected under 675 nm laser excitation [pink trace] and upon co-excitation with the 675 nm laser (100 μW) and the 532 nm laser, the power of which was varied (purple, cyan, ochre and green traces) as emphasized in the figure by the green arrow. The dip in $g^{(2)}(\tau)$ at short (ns) time scales confirms that the emitter is a quantum emitter with sub-Poissonian statistics. The exponential decays at longer (ms) time scales reveal the presence of additional intermediate/metastable levels. The best fit to the data are achieved using a four-level model (Fig. 2c), where two of the exponential decays observable in Fig. 2a correspond to states whose photo-dynamics is non-trivial, as discussed below.

The time constants ($\tau_1$ and $\tau_2$) for these two intermediate/ metastable levels—as obtained from the fit—are plotted in Fig. 2b (cf. Methods). Interestingly, the two-laser co-excitation measurements show that these states are depopulated by the addition of the 532 nm laser, even at a very low power, ~ 0.1 μW. The relative time constants $\tau_1$ and $\tau_2$ indeed decrease (Fig. 2b) for increasing excitation powers of the 532 nm laser (in the investigated range ~ 0.1–50 μW). This decrease is nonlinear and it correlates consistently with the enhancement in PL intensity observed in Fig. 1e. Simultaneously—and remarkably—the addition of the 532 nm laser affects the saturation behaviour of the emitter. Specifically, repumping via the 532 nm laser reduces the intensity required to saturate the emitter. This is highlighted in Fig. 2d where the saturation curve upon sole 708 nm laser excitation (purple trace) is compared with two curves obtained upon co-excitation with the 708 nm and the 532 nm laser pair, with 532 nm laser powers being 1 and 10 μW (blue and green traces). In brief, under 708 nm excitation the emitter saturates at ~ 14 mW. Under co-excitation, the emitter saturates at ~ 3 mW ($P_{532nm} = 1$ μW) and ~ 1.5 mW ($P_{532nm} = 10$ μW), respectively— that is, the use of the laser pair reduces the saturation power by approximately one order of magnitude. In addition, the emitter shows a peculiar fluorescence intermittency (blinking) behaviour, which is dependent, as well, on the 532 nm excitation. Figure 2e–g summarizes the main characteristics. Under excitation with the sole 708 nm laser, the emitter exhibits two fluorescent thresholds (Fig. 2e, left) with the statistics for the photon distribution (Fig. 2e, right) showing the system to be mainly in the lower fluorescence one. It is noteworthy that there appears not to be an off-threshold, which would correspond to a completely dark state

with the number of detected photocounts equal to the background level. By adding the 532 nm laser of increasing power, the statistic of the photon distribution shifts towards the threshold of higher fluorescence (Fig. 2f,g).

A possible model consistent with all our observation is presented in Fig. 2c. The long autocorrelation measurements show that the emitter essentially behaves similar to a four-level system with a fast-decaying intermediate and a long-lived metastable state (see below). Off-resonant excitation with either a 532 nm or a 675 nm (or 708 nm) laser leads to emission into the ZPL, with relative intensities in excellent agreement with recent reports[22]. Addition of the 532 nm to the 675 nm excitation laser suppresses the population of the intermediate state via repumping to the excited state(s), which results in the observed enhanced, nonlinear photoemission. This repumping effect is non-trivial: the nonlinear increase in PL is accompanied by a reduction in the intensity required to saturate the emitter (Fig. 2d), which indicates the existence of a complex dynamics involving both the intermediate and the metastable states. Our interpretation is that upon sole 675 nm (or 708 nm) excitation the system can undergo a transition to a dark state through the long-lived metastable state, with this dark state possibly being a different charge state of the emitter. This is consistent with the blinking data in Fig. 2e, where we see the emitter's photo-statistics being mainly at the low-fluorescence threshold upon sole 675 nm excitation. As a caveat, we deliberately use the word 'threshold' rather than 'state', as the binning average is oblivious to transitions of fast dynamics, which is potentially the case in the hypothesis of transitions to a different charge state. It is noteworthy that this is also consistent with the fact that we do not measure a non-fluorescent off-state in the blinking trace, indicating that we are indeed averaging over very fast-dynamics transitions[23]. It also allows us to weaken the possibility that these transitions occur towards trap states with slow dynamics (as these would be detected as a complete off-state in the fluorescence time trajectory). The blinking analysis also shows (Fig. 2f,g) that the addition of the 532 nm repumping laser suppresses the probability for the emitter to be in the low-fluorescence threshold and allows us to infer about the relative dynamics between the fast-decaying intermediate and long-lived metastable state. Specifically, the repumping via the 532 nm laser reverts the system from the intermediate state to the excited state(s), thus inhibiting the otherwise faster non-radiative decay from the intermediate to the ground state, which would occur without repumping. This efficiently depopulates the ground state (cf. Methods) resulting in the observed reduction in excitation intensity required to saturate the emitter.

A detailed modelling and analysis of the photo-kinetics of the centre and its dependence on the 532 nm laser repumping is presented in the Methods section of the paper and is shown in the Supplementary Note 2 and Supplementary Figures 3–5. As mentioned before, the same behaviour could be reproduced on an entire subclass of emitters with emission ZPLs > 700 nm (cf. Supplementary Figures 1–2).

**Fluorescence nanoscopy with emitters in hBN**. As a demonstrative application, we show that the unique photo-physical properties of this class of hBN quantum emitters can be harnessed to realize a new modality of far-field, sub-diffraction fluorescence nanoscopy. Super-resolution nanoscopy methods grouped under the umbrella of Reversible Saturable Optical Fluorescence Transitions (RESOLFT) techniques rely on two key criteria: the first one is the existence of controllable bright/dark-like transitions between the emitters' states. The second criterion

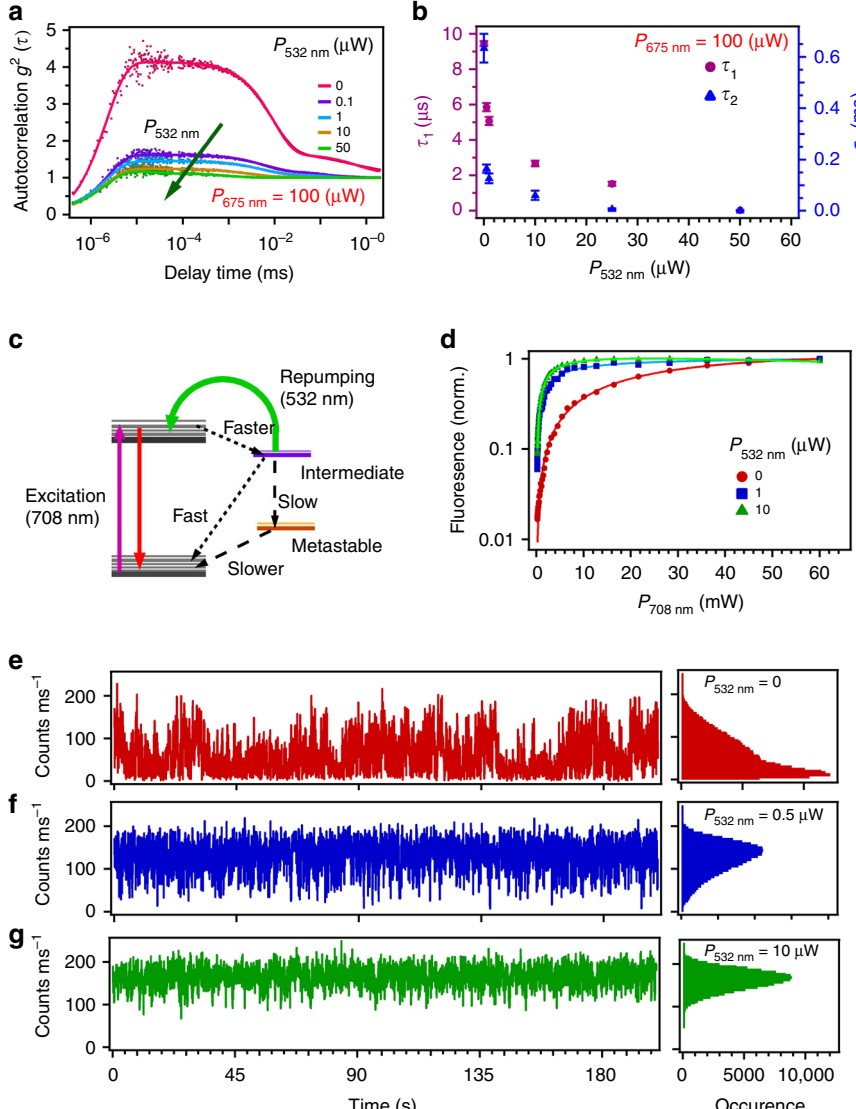

**Fig. 2** Photo-physics of the single emitter introduced in Fig. 1. **a** Long autocorrelation measurements under excitation with a 675 nm laser (100 µW). Increasing the power of a coincident laser (532 nm, green arrow) suppresses the population of the intermediate/metastable states due to repumping of the emitter (see main text). The experimental data (points) are fitted (solid lines) using a four-level model. Error bars have been calculated based on the fitting to the autocorrelation data. **b** Reduction in the time constants $\tau_1$ and $\tau_2$ associated with the intermediate/metastable states (extracted from the fits in **a**; see Discussion in the main text) caused by increasing the power of the repumping 532 nm laser. **c** Simplified level structure of the emitter (see main text). The emitter possesses a ground state and excited state(s), as well as a fast-decaying intermediate and a long-lived metastable state. Radiative transitions are indicated by straight, solid arrows, repumping via the 532 nm laser is indicated by the green arrow, and fast and slow non-radiative transitions are indicated by dashed arrows. **d** The repumping causes a reduction in the laser power that is needed to saturate the emitter. **e**, **f** Fluorescence time trajectories of the emitter sampled into 100 ms bins, under excitation with 708 nm **e** or co-excitation with 708 and 532 nm lasers **f**, **g**. The power of the 708 nm laser was kept at 100 µW and the power of the 532 nm repumping laser was 0.5 and 10 µW in **f** and **g**, respectively. The corresponding histogram of the photon distribution at each excitation condition is shown on the right

is the resilience of the emitters (and the enclosing environment) against high-power excitation, as the transition to the dark state is typically achieved via optical saturation. For instance, stimulated emission depletion (STED)[24] and ground-state depletion (GSD)[25] methods exploit spatially modulated (e.g., doughnut-shaped) light beams with ultrasharp bright/dark gradients to selectively image emitters. In STED techniques, this is achieved by 'switching off' the emitters around the doughnut null (where the emitter to be imaged is located) by inducing stimulated emission (at saturation) between the first optically excited ($S_1$) and the ground ($S_0$) states. In GSD, the same outcome is achieved by shelving the system to a metastable, long-lived dark-state ($T_1$, usually accessible from $S_1$).

The class of hBN emitters presented in this work satisfy both criteria. We therefore used them to realize GSD imaging using the experimental setup shown in Fig. 3a. Briefly, a 532 nm and a 708 nm laser are co-aligned and simultaneously focused through an aberration-corrected objective (numerical aperture, (NA) = 0.95). The photons from the emitters under investigation are back-collected with the same objective, focused into the aperture of an optical fibre (used as a confocal-microscope pinhole) and directed to an avalanche photodetector (cf. Methods). Depending on the excitation scheme (see below), we employed vortex phase masks to modulate both the 532 nm and the 708 nm excitation laser into having a spatial doughnut-shaped profile, with a near-zero intensity in the centre (Fig. 3a, inset).

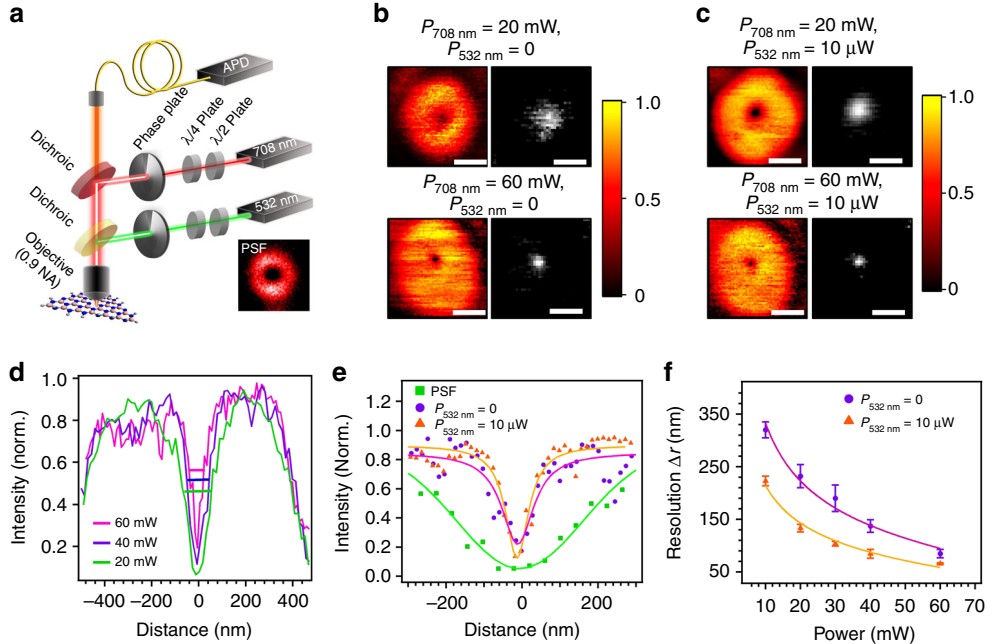

**Fig. 3** Super-resolution nanoscopy of the emitter shown in Fig. 1 performed using a single laser and a coincident laser pair as the excitation source. **a** Schematic of the setup used to perform GSD nanoscopy employing excitation lasers with doughnut-shaped intensity profiles. The system point spread function (PSF) was determined by reflection of the lasers from a 50 nm gold nanosphere (inset, red circle). **b** Negative GSD images of the single defect under excitation with a single 708 nm, doughnut-shaped laser and using laser powers of 20 mW and 60 mW as indicated. The direct GSD images on the right are obtained by linear deconvolution of the negative GSD images on the left. **c** Negative GSD images of the same emitter after addition of a 10 μW, 532 nm, doughnut-shaped repumping laser co-aligned with the 708 nm laser. The images on the right are linear deconvolutions of the negative ones. **d** Photoluminescence intensity profiles of negative GSD in **b** showing a resolution improvement at higher excitation powers. Solid lines indicate the full-width at half-maximum (i.e., the resolution) of the emission null at the centre of the doughnut. **e** Intensity profiles used to compare the resolution obtained from negative GSD performed at 40 mW of 708 nm laser using the single doughnut beam (circles, violet) and the co-incident laser pair (triangles, orange). For comparison, the intensity profile obtained from reflection of 50 nm gold nanoparticles (squares, green) is shown as point spread function of our setup. **f** Dependence of GSD resolution on the power of the 708 nm laser, with and without the co-incident 532 nm, 10 μW repumping laser. Scale bars in **b** and **c** are 300 nm. Error bars have been calculated based on repeating the measurement for three times

We start by performing standard negative GSD nanoscopy using the 708 nm doughnut-shaped laser as the excitation source (Fig. 3b). Negative GSD[26] is possible with this class of hBN emitters as our photo-kinetics analysis revealed they possess at least one long-lived metastable dark-state and they are photo-stable at high excitation powers. As the beam is scanned, the emitter experiences a doughnut-shaped excitation intensity profile, which produces a corresponding high–null–high emission pattern. With this configuration, in the confocal image the emitter's location coincides with the centre of the emission null. Sub-diffraction resolution is achieved as at higher powers of the scanning doughnut beam the high–null and null–high PL emission gradients become steeper, which effectively narrows the full-width at half-maximum (FWHM) of the emission null. The minimum in intensity yields an inverse image of the emitter with a spatial resolution that exceeds the diffraction limit. The mathematical deconvolution[27] (cf. Methods) of the negative GSD image (Fig. 3b, left) yields a direct GSD image of the emitter (Fig. 3b, right).

Figure 3d shows the GSD resolution (extracted from the FWHM of the null) that we can obtain by varying the power of the 708 nm doughnut-shaped laser. With the experimental parameters of our setup (NA = 0.95, $\lambda_{Exc}$ = 708 nm), we reach a resolution of (87 ± 10) nm at 60 mW, well below ~ 460 nm, which is the diffraction-limited resolution measured for our confocal setup. The resolution $\Delta r$ is given by:[26]

$$\Delta r \cong \lambda (\beta \pi n)^{-1} \sqrt{\varepsilon + \frac{I_s}{I_m}} \qquad (1)$$

where $I_m$ is the maximum laser intensity in the crest of the doughnut, whereas $\epsilon I_m$ is the minimum (null) intensity in the centre. The quantity $n$ is the refractive index of the medium and $I_s$ is the laser intensity at which the emission intensity equals half of the maximum value in the limit of infinite excitation power. The parameter $\beta$ is the steepness of the point spread function (PSF) and depends both on the emitter properties and the crest-to-minimum intensity gradient of the doughnut-shaped excitation source. For our purposes, it is hereby relevant to point out that GSD resolution is, in principle, non-diffraction-limited: it improves by increasing the excitation laser power beyond $I_s$, so as to minimize the ratio $I_s/I_m$. This usually translates in the need to use high laser powers to achieve high spatial resolution, which is the main drawback of GSD methods and the related suite of RESOLFT imaging techniques[27]. The high excitation powers needed to break the diffraction limit usually result in bleaching of most emitters and thus restrict the robust use of RESOLFT methods to a limited number of systems such as the highly photostable colour centres in diamond[28]. High excitation powers are undesirable for another reason: they induce heating and can damage the surrounding environment, which is particularly problematic for bio-imaging nanoscopy applications.

Remarkably, these problems are alleviated by the unique photo-physics of the class of hBN emitters presented here. With reference to Equation 1, if $I_s$ is reduced, the resulting decrease in the ratio $I_s/I_m$ leads to an improvement of the GSD image resolution whilst maintaining a fixed laser power (i.e., a fixed value of $I_m$ in Equation 1). The ability to reduce $I_s$ in our system is evident from the PL saturation curves shown in Fig. 2c: the co-

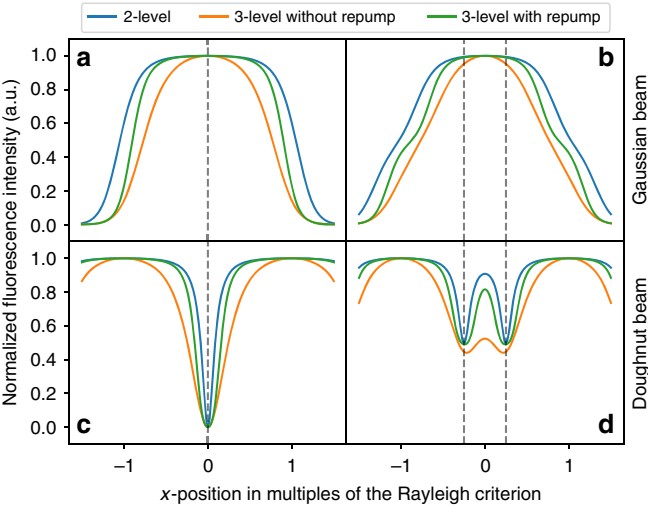

**Fig. 4** Calculation of the fluorescence profile for the emitter under different laser excitations. **a**, **b** Cross-section profile of a single emitter **a** and two closely positioned emitters **b** excited with a Gaussian beam. **c**, **d** Cross-section profile of a single emitter **c** and two closely positioned emitters **d** excited with a doughnut beam. The blue (orange) lines correspond to emitters treated as two- (three-) level systems. The green line refers to the case where a second repumping laser is added—with Gaussian **a**, **b** and doughnut **c**, **d** profile, respectively. In **a–d**, the *x* axis represents the Rayleigh criterion so that the distance between max and min of the doughnut is 1. The vertical dashed lines indicate the position of the emitter along the *x* axis

excitation of the emitter with a 708 nm laser plus a low-power 532 nm repumping laser causes $I_s$ to decrease. Based on this observation, we thus expect the GSD resolution to improve if the imaging is performed using a coincident pair of excitation lasers. To verify this, we perform GSD with a low-power (10 μW), doughnut-shaped 532 nm repumping laser co-aligned with the 708 nm main doughnut beam. The addition of the repumping beam indeed produces higher resolution images of the emitter, as shown in Fig. 3c, right (the corresponding direct images were again obtained via linear deconvolution; Fig. 3c, left). The improvement is detailed in Fig. 3e, which compares the normalized intensity profile of the emitter excited by a 40 mW, 708 nm beam before (violet circles) and after (orange triangles) applying the 10 μW, repumping 532 nm laser, as well as the PSF of our conventional confocal setup (green squares), obtained from the reflection image of a 50 nm gold nanoparticle.

A plot of GSD image resolution vs. power—up to the maximum power achievable with our experimental setup—is shown in Fig. 3f for both the single (708 nm doughnut) excitation laser and the laser pair (708 nm doughnut + 532 nm doughnut). The highest resolutions we achieve using the one and the two-laser excitation schemes are (87 ± 10) and (63 ± 4) nm, respectively. Noticeably, the power of the 708 nm laser needed to achieve a given target resolution is improved dramatically by the addition of the low-power (10 μW) 532 nm repumping laser. For reference, a resolution of 100 nm is achieved with 55 mW of excitation power when using the single doughnut vs. the 30 mW (+ 10 μW of the 532 nm laser) of the laser pair. This reduction in laser power is highly desirable for super-resolution nanoscopy as it mitigates the power-induced heating and damage of the sample. Interestingly, there is no intrinsic limitation to extending this repumping scheme to other quantum emitters, provided they possess analogous nonlinear photo-kinetics. This could potentially spur the advancement of novel, alternative schemes for super-resolution imaging.

We note that current implementations of GSD imaging based on dual-beam schemes employ a coincident pair of a doughnut plus a Gaussian-shaped laser[26,29]. This configuration is inappropriate here: the low-power 532 nm laser is used to reduce $I_s$ and this effect must be maximized in the crest of the doughnut-shaped, high-power 708 nm beam, while maintaining an intensity null at the beam axis (i.e., in the centre of the doughnut).

To further understand/enhance the effects of repumping on the obtained super-resolution images, we calculate the fluorescence profiles of the emitter for different beam shape configurations (Fig. 4). For completeness, both the case of a single and that of two closely positioned emitters are considered. The graph displays the calculated fluorescence of the quantum emitters considered as two-level systems (blue trace) and three-level systems with (green trace) and without (orange trace) the additional laser repumping. Figure 4a,b show that the excitation of the emitters with two superimposed Gaussian beams of different wavelength, does not achieve resolutions beyond the diffraction limit. Conversely, when doughnut-shaped beams are employed, super-resolution can be obtained via negative GSD. Figure 4c,d show that the co-excitation of the emitters with two doughnut shaped beams (one employed for repumping purposes, as described above) results indeed in improved resolutions. Our calculations provide a qualitative match to the experimental results (Fig. 3d–f).

Interestingly, our calculation indicates that super-resolution imaging with a two-level system is also possible. However, this requires operating at saturation, which, in practice, is only feasible under resonant excitation at cryogenic temperatures[30]. On the other hand, we show that exploiting the level scheme and introducing a repumping laser, allow for super-resolution imaging at significantly lower laser excitation powers, at room temperature.

## Discussion

We report the existence of a class of hBN quantum emitters with a highly nonlinear optical behaviour. The emitters possess a fast-decaying intermediate and a long-lived metastable state accessible from the first excited electronic state and optically reversible to the excited state(s) by means of a two-laser repumping scheme. This results in a nonlinear photo-emission behaviour of these emitters which produces enhanced PL and reduced saturation intensity. Beyond the intriguing photo-physics of these emitters, we demonstrate that their unique properties can be harnessed to realize a new modality of far-field, super-resolution imaging with a dual-doughnut-beam configuration. We report sub-diffraction resolution of ~ (63 ± 4) nm. We envision that this technique could be extended to other stable single emitters in one-, two- and three-dimensional material hosts, as well as other fluorophores used for super-resolution nanoscopy. The nonlinear behaviour and the repumping mechanism can also be used to suppress spectral diffusion, and thus aid with the generation of indistinguishable photons from single photon emitters in hBN. The impact of our findings is twofold. They deepen the current understanding of the photo-physics of quantum emitters in layered hBN, as well as show new potential ways of harnessing their nonlinear optical properties for specific applications such as the hereby presented sub-diffraction nanoscopy. During the preparation of the manuscript, we became aware of a related work from Radenovic and colleagues[31] that exploits blinking behaviour of localized emitters in hBN.

## Methods

**Sample preparation.** hBN flakes (Graphene Supermarket) were dropcast onto a silicon substrate and annealed at 850 °C in Ar, in order to activate the emitters[9,11,16].

**Confocal and GSD microscopy**. PL measurements were carried out in a home-built confocal setup. The sample was mounted onto a XYZ piezo stage (Physik Instrumente-Nanocube P-611) with positioning resolution of 0.2 nm. Optical excitation was performed using different laser sources as follows: 675 nm laser (PiL051XTM, Advanced Laser Diode Systems GmbH)—used for PL and autocorrelation measurements; a 532 nm laser (Shanghi Dreamlasers, 532 nm low-noise CW laser)—used as repumping laser for PL, autocorrelation and saturation measurements; Ti:Saph (M-squared, 700–750 nm)—used for saturation measurements, studying the effect of excitation wavelength and super-resolution imaging; and Supercontinuum (NKT Photonics, Fianium WhiteLase supercontinuum laser) equipped with Acousto-optic Tunable Filter (400–550 nm)—used for studying the effect of repumping.

The power of the lasers was measured in front of the objective. To make a doughnut-shaped beam, the laser was first linearly polarized via a polarizing beam splitter (Thorlabs AR-coated Cube beam splitter) and then passed through a zero-order half and a quarter phase plate (Thorlabs-Zero order waveplates) to achieve circular polarization. The beam was then directed through a vortex phase mask (RPC Photonics, VPP-1b for 708 nm and VPP-1c for 532 nm laser). The 708 nm and 532 nm lasers were guided to the sample using a long-pass filter (Semrock 785 nm EdgeBasic) and a dichroic mirror (Semrock 532 nm dichroic), respectively, and were focused on the sample through an aberration-corrected objective lens (Nikon 1× 100, NA = 0.95). The emission collected from the same objective was filtered using a notch filter (Semrock, 785 nm StopLin notch filter) and a 780 nm long-pass filter (Thorlabs, long-pass colour filter), and then coupled to the fibre that was connected to a spectrometer (Acton Spectra ProTM, Princeton Instrument, Inc.) equipped with a 300 lines per mm grating and a charge-coupled device detector with a resolution of 0.14 nm, or splitted into 50:50 in a HBT interferometer for autocorrelation measurement using two avalanche photon detectors (SPCM-AQRH-14-FC, Excelitas Technologies TM) and a time-correlated counting module (Picoharp300TM, PicoQuantTM).

**Deconvolution**. To retrieve the direct image from the negative GSD scan, linear deconvolution was applied using built-in functions in Matlab. First, the high-resolution details (mainly the centre local minimum in the negative GSD image) was removed using a short-pass Gaussian filter to produce a blurred image. Next, the direct image was extracted by subtracting the GSD image from the blurred image. The blurred image from the application of the short-pass Gaussian filter (mathematical) method is preferred over the normal confocal scan of the image because of possible mismatch between confocal image and GSD image due to drifting during data acquisition. It is otherwise still possible to use the confocal map instead of the mathematical method[27].

**Autocorrelation data**. Autocorrelation data in Fig. 2a was fitted with the following equation:

$$g^2(\tau) = 1 - (1 + a_1 + a_2)\exp\left(\frac{\tau}{\tau_{exc}}\right) + a_1\exp\left(\frac{\tau}{\tau_1}\right) + a_2\exp\left(\frac{\tau}{\tau_2}\right) \quad (2)$$

In the four-level model we used (cf. main text), three decay rates are considered for the excited state and two intermediate/metastable (dark) states. The bunching time that corresponds to the excited state decay is shown in Fig. S5.

**Modelling of the fluorescence intensity profile**. The model is based on a set of rate equations that describe the time evolution of a quantum emitter electronic state, with the four-level scheme shown in Fig. 2c. It consists of a ground state $|g\rangle$, an excited state $|e\rangle$, a fast decaying intermediate state $|i\rangle$ and a long-lived metastable state $|m\rangle$, which are connected via rates $k_{ij}$, with $i$ and $j$ being the levels to which the rates refer.

$$\dot{n}_g(t) = -k_{ge}n_g(t) + k_{eg}n_e(t) + k_{ig}n_i(t) + k_{mg}n_m(t)$$

$$\dot{n}_e(t) = k_{ge}n_g(t) - (k_{eg} + k_{ei})n_e(t) + k_{ie}n_i(t)$$

$$\dot{n}_i(t) = n_e(t) - (k_{ig} + k_{ie} + k_{im})n_i(t)$$

$$\dot{n}_m(t) = k_{im}n_i(t) - k_{mg}n_m(t) \quad (3)$$

Here, $n_i$ is the time dependent population of each state. A repump mechanism from the intermediate to the excited state is motivated by the measurements shown in Fig. 2 and reflected in the model via the transition rate $k_{ie}$. This repump rate is assumed (see below) to be proportional to the 532 nm laser intensity, whereas the rate excitation rate $k_{ge}$ is assumed to be proportional to the 708 nm laser. Under continuous wave illumination, the system reaches the steady state $(\dot{n}_g(t) = \dot{n}_e(t) = \dot{n}_i(t) = \dot{n}_m(t) = 0)$ and the set of rate equation 2 can be solved. This gives the following average probability of the system to be in the excited state

under continuous excitation:

$$n_e = \frac{k_{ge}k_{ig}(k_{me} + k_{mg} + k_{mi})}{k_{em}k_{mi}(k_{ge} + k_{ig}) + k_{ig}(k_{em}(k_{ge} + k_{mg}) + (k_{eg} + k_{ge})(k_{me} + k_{mg} + k_{mi}))} \quad (4)$$

Multiplying Equation 4 by the radiative decay rate $k_{eg}$ provides a direct relation between the contributing rates and the maximum detectable fluorescence intensity. Based on the measurements shown in Fig. 2, it is justifiable to assume that the transition from the first excited optical state to the intermediate state, and the subsequent relaxation towards the ground state are on a much shorter time scale than the radiative transition from the excited to the ground state. However, the intermediate state also provides a decay channel to a long-lived metastable state (possibly a different charge state of the emitter, see main text). Scanning an excitation or repump beam across a fixed emitter changes the corresponding transition rate proportionally to the local beam intensity. While keeping the rates of all relaxation channels constant ($k_{eg} = 1$, $k_{ei} = 10$, $k_{ig} = 100$, $k_{im} = 0.1$, $k_{mg} = 1$), we can qualitatively model the expected fluorescence intensity profile for the different scanning beam shapes and excitation/repump schemes presented in Fig. 4. Analogously to Ref. [24], we construct the doughnut beam intensity profile from two Airy functions, spaced apart so that their first minima perfectly overlap. In Fig. 4 the distance between one maximum and the centre of the doughnut represents the Rayleigh criterion and is the scale of the $x$-axis. The Gaussian beam root mean square spot size was chosen to be 0.42: this is the length to approximate the shape of a single Airy profile. The intensities were scaled such that the maximum repump rate was $k_{ie,max} = 1000$, indicating a highly efficient repump mechanism, and the maximum excitation rate was $k_{ge,max} = 100$, in order to reach saturation at the beam profile maxima.

**Data availability**. The data that support the findings of this study are available from the corresponding author upon request.

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

## Acknowledgements

We acknowledge the financial support from the Australian Research Council (DE130100592 and DP140102721), FEI Company, the Asian Office of Aerospace Research and Development grant (FA2386-17-1-4064) and Office of Naval Research Global (N62909-18-1-2025). This research is supported in part by an Australian Government Research Training Program (RTP) Scholarship. B.S. acknowledges the support by the Deutsche Forschungsgemeinschaft, DFG, (project C2 in the CRC951).

## Author contributions

M.K., C.B., B.S. and A.W.S. conceived and designed the experiments. M.K. and C.B. performed the experiments. B.S. performed the calculations. M.N., T.T.T., Z.X., F.W., S. K., A.W.S and B.S. assisted with analyzing data. C.B., I.A. and M.T. supervised the project. M.K., C.B., I.A. and M.T. wrote the manuscript. All authors discussed the results and commented on the manuscript.
