## [Peer Review File · Nature Communications]

Reviewers' comments:

Reviewer #1 (Remarks to the Author):

In this work Kianinia et al. demonstrate two major results: they show that quantum emitters in hBN are sufficiently photostable to achieve superresolution imaging. To obtain this results, they exploit the low power modality of GDS nanoscopy. In particular low power excitation of (10 μ W), doughnutshaped, green laser is used in combination with the standard high power (tens of mW) excitation doughnut to achieve GSD imaging. Authors report a spatial resolution of $\sim(63\pm 4)$ nm. Authors also present a comprehensive characterization of this unique single photon emitters and provide spectral, polarization and antibunching. In addition, to better understand the photophysics of the emitters they characterize the photophysics by varying the intensity of the green laser.). Increasing the power of a 532 laser results in the suppression of the population of the metastable states due to repumping of the emitter. These measurements allowed authors to suggest a simplified level structure for the emitter. The experiments are well done, interesting, worthy of further study, and likely to be of broad interest to the readers. I support publication after my minor comments are addressed.

1.

In the work authors used liquid exfoliated hBN flakes from Graphene supermarket, given the small size of monolayer flakes, authors should provide more details on the methods used to identify single layers.

2.

On page 2 authors discuss the requirement for the photostability of the probes at RT and that this property is very rare –I would disagree here with the authors, for example, lanthanide-doped upconversion nanoparticles (UCNPs) have become attractive STED imaging probes, <https://www.nature.com/articles/nature21366> while FNDs are well established samples for nanoscopy <https://www.nature.com/articles/nphoton.2009.2>. Both classes of nanoparticles are bright and stable photo-emitters

3.

Authors should better discuss the advantages and novelty of their low power modality of GDS and compare it with the standard STED imaging (107 W/cm²) RESOLFT, and MINIFLUX modalities

Reviewer #2 (Remarks to the Author):

Super-resolution imaging of quantum emitters in layered materials by Mehran Kianinia et al.

The authors present a photo-physical study of hBN samples, focusing partially on their photoluminescence and primarily on an optical scheme to enhance the spatial resolution in confocal imaging of the same photoluminescence. The resolution enhancement is based on the RESOLFT concept (seeking reversible saturation in fluorescence), originally developed by S. Hell. The manuscript reads very well and the data are convincing, although I disagree on some minor technical points in the data analysis (see below).

With respect to the breakthrough level and general interest of the work, in view of the large amount of work in super-resolution available in the literature, I am not convinced that the manuscript reveals a breakthrough in super-resolution but the manuscript will certainly be of interest to the super-resolution community. Conversely, when considering the photo-physical properties of hBN defects, there are potentially new discoveries presented here, but I am not well positioned to judge these. Finally, when considering the need for super-resolution to advance the understanding of hBN for future applications of the materials, it is also not clear if the present manuscript is a breakthrough. Indeed, the paper quoted ref. 33 reveals 10 nm resolution separating features for a similar hBN sample, which is better than in the present manuscript (ca.

60-70nm; and without truly resolving/separating adjacent features, the quoted value in the present manuscript corresponds to a single feature fwhm in the image, which is not necessarily a good proof of resolution (C Sheppard)). Overall, thus I feel that the authors do not sufficiently position their results apart/above from the literature in super-resolution to justify a breakthrough in super-resolution, but the authors present a very interesting story nonetheless.

I have written some points that support the build-up of my opinion above.

- 1) It is clear that hBN is an interesting material and well worth of studies and in this aspect the manuscript is very rich. On the other hand, the orientation given to the manuscript to propose the use of hBN as label for super-resolution (eg, line38-39 etc) is probably premature. First, there are large variations in the photoluminescence spectra recorded across different parts of the sample and thus a large variety in the defect natures. Very few of these appear suitable to truly gain from the high resolution "repumping (@532nm)" imaging discussed. Thus to be usable as label, more must be known on the emitting defects and more must be known on how to prepare these. Targeted studies should be made (I assume these would also involve TEM for example). Second, it is clear that for some few defects the addition of the low power beam at 532 nm truly helps to reach saturation but this still requires tens of mW in the primary beam which is not really advantageous versus current STED labels for example. Third the resolution is at the end not better than STED. Finally, how would "pieces" of hBN be attached to detection proteins or other biomolecules (without altering the photo-physical properties of the hBN defects involved)?
- 2) It is unknown whether the scheme can be generalised to other materials (ie, adding the low power green to reach saturation with less primary power). It must be pointed out as well, that the literature contains already many works where variations of RESOLFT have been proposed/demonstrated and where the typical STED (or here GSD) beam scheme is modified to probe various materials including non-fluorescent ones (STED/PALM and derivative methods are already covering very well the fluorescence).
- 3) I find the photoluminescence analysis to be very well done throughout the paper. It is perhaps needed to indicate the temporal structure of all the lasers used. The list of laser sources is quite extensive and it is unclear which laser is used to record a given datasets. This would be helpful for readers and rather easy to add.
- 4) I have no doubt that the resolution limit is "broken" in this manuscript. Yet, it is essentially achieved via the GSD scheme introduced by S. Hell a decade or so ago. The primary doughnut excitation images show this clearly and is responsible for most of the super-resolution. The novelty lies in the addition of the weakly powered 532 nm doughnut. Although the effect is impressive for some of the defects that are probed, the overall gain in terms of resolution is relatively minor at the end, since the system seems to be quite photo-stable anyway (basically the order of magnitude of powers are still the same as for standard GSD). Thus at the end one has a GSD scheme as defined by S. Hell with a defect-dependant power economy. Thus also the idea of achieving super-resolution at lower power is overall desirable, it is unclear if it is achieved on a materials that effectively needs it here. Fig3f shows that the resolution with and without the additional doughnut are essentially converging (when increasing the primary power) experimentally to a same value. It would be interesting for the authors to consider discussing what would happen if the primary power were to be increased further (would the 2 curves effectively exactly match?)
- 5) In Fig2d, the saturation curve for 10 microW seems to decrease at higher 708nm power? Why is that? This could be detrimental to the super-resolution should the resolution be pushed further.
- 6) Line171 and methods. The authors use a "deconvolution". The methodology described seems to be a high pass spatial frequency filter (ie, the low frequencies are removed when subtracting the smoothed doughnut image). It is unclear that this method would work in general (in fact, the simulations in Fig4d suggest that artefacts will appear already with 2 identical objects). The operation here seems to be a differentiation (thus analogous to S. Hell's GSD scheme) but with low pass filtered doughnut image instead of a Gaussian-PSF image. The subtraction is not a bad operation here (if one is careful to the potential formation of sidelobes in the subtracted image where the doughnut and reference (usually Gaussian) do not match well at the edges. There are

subtraction algorithms that have appeared recently that should handle this issue. As currently used here, it is unclear that the "deconvolution" here would work well on large image involving more than one emitter. It is also not fully clear that the operation eventually leads to a proper deconvolution. In my view, the "deconvolution" used here would have to be seriously "upgraded" for the scheme to work on specimens that involve more than very sparse emitters. How would the method fare for an image such as the one presented in the first figure in the SI?

7) Line 212-213: "The highest resolution that we measured using the one- and two-laser excitation schemes is (63 ± 4) nm and (87 ± 10) nm, respectively". This is probably the other way around.

8) Fig 4: There is very little detail on the modelling of the line profile shown and little detail about the parameters involved (power? beam fwhms? wavelengths? relaxation times? how close are the emitters? etc), perhaps more could be written into the SI. Moreover when looking at Fig 4d, it is not obvious to say that the "repumping" affords a better resolution than without it? (line 253-254). Both emitters are quite distinct in all the cases shown (and all dip half-way more or less, in both cases). Maybe the statement line 253-254 and line 270-271 should be clarified and providing more info on the simulations may help with that. Just to be clear, I agree with the statement line 270-271 but I am not convinced that this is shown in the simulations. Thus the simulations do not appear to be truly supporting the manuscript.

9) Methods: Sample preparation: the authors probably mean hBN and not graphene?

Reviewer #3 (Remarks to the Author):

In the manuscript titled "Super-resolution imaging of quantum emitters in layered materials" the authors (Kianinia et al.) report ~ 60 -nm spatial (lateral) resolution in locating quantum emitters in hexagonal boron nitride. To achieve this resolution, the authors employ one- and two-laser schemes for ground state depletion and demonstrate the stability of the hBN quantum emitters for potential applications in label based super-imaging.

The experimental results presented in the first three figures are reliable and consistent. The quantum nature of these emitters is supported with a long time scale pumping study in the presence of an additional excitation laser in order to reveal the internal structure of these debated systems. Figure 3 is the punchline of the manuscript and displays the scaling of the spatial resolution the authors achieve dependent on the lasers used. Figure 4 is a simulation plot to present an understanding of the internal level scheme of these quantum emitters using super-resolution results.

The manuscript results are of sufficient quality and interest to warrant publication in Nature Communications, but I have a few remarks to improve the readability and the clarity of the message:

1) The sentence starting with Furthermore in the introductory paragraph is not clear. I now know what they wanted to say after having read the manuscript, but this sentence confuses the reader. The nonlinear photophysical property is perhaps more indicative of a nonlinear optical response rather than a saturation-based effect.

2) Line 56: Is there a reason why the authors use quantum emitters and SPEs separately in the same sentence? It sounds like they are different entities and photostability of one affects the resolution of the other.

3) Line 64: The authors use (in many other parts too) phrases like "very high" or "low" and it is not clear from the manuscript to what these refer. For example are they in the end showing an order of magnitude reduction in the needed laser power, or just a factor 2?

4) Figure 1 panel c: The authors do not mention the solid curve (fit) to the data. Is there a background in the intensity? It might be good to mention the fitting function.

5) Figure 1 panel d: Why do the authors pick 10 μ W and 300 μ W for the two lasers? Are these special values (i.e. saturation etc)

6) Figure 1 panel e: I have a hard time seeing what the authors want me to see here. The image

gets brighter by laser power, but the nonlinearity is far from obvious. I suggest an intensity plot like panel c and an inset image of the spatial resolution as an example.

7) Figure 2 panels a and b: Isn't it possible to extract from the autocorrelation measurements a power-dependent τ_1 to compare with panel b? The dynamics should reveal the τ_1 one expects and contrast it to the measured data.

8) Finally, from my perspective, the manuscript offers a new technique for superresolution using a class of emitters matching the internal level scheme of the hBN quantum emitters. This is the bigger fish, the fact that we can locate the quantum emitters with better resolution is less important. The main reason for this is that imaging these quantum emitters with 60-80 nm resolution as a main result has less appeal because this is not a particularly attractive length scale to either reveal the internal structure of the quantum emitters or have superior localisation in a photonic chip. Simple high-quality confocal imaging and inference allows for $\sim 20-40$ nm localisation already, so the authors should at least beat that if they want this to be their main result. My main suggestion would be a revision of the manuscript to put the new technique more in the spotlight.

Reviewer #1

1) In the work authors used liquid exfoliated hBN flakes from Graphene supermarket, given the small size of the monolayer flakes, authors should provide more details on the methods used to identify single layers.

The liquid exfoliated hBN flake used in this work are 'as-received' from the supplier: *Graphene Supermarket*. Here is the link to the actual webpage containing the specifications of the material: <https://graphene-supermarket.com/Boron-Nitride-Pristine-Flakes-in-Solution.html>.

They are multilayered hBN flakes and they have not undergone any further treatment except for the Ar-annealing process at 850 °C discussed in the Methods section of the paper. A thorough characterization of the hBN flakes including atomic force microscopy (AFM) and scanning electron microscopy (SEM) analyses can be found in previous publications both from our group (Kianinia, M., et al. "Robust Solid-State Quantum System Operating at 800 K." *ACS Photonics* 4.4, 2017, 768-773) and other research groups (Jungwirth, N. R., et al. "Temperature dependence of wavelength selectable zero-phonon emission from single defects in hexagonal boron nitride." *Nano letters* 16.10, 2016, 6052). For completeness and clarity, we have added the relevant references in the Method section of the paper.

2) On page 2 authors discuss the requirement for the photo-stability of the probes at RT and that this property is very rare –I would disagree here with the authors, for example, lanthanide-doped upconversion nanoparticles (UCNPs) have become attractive STED imaging probes, <https://www.nature.com/articles/nature21366> while FNDs are well-established samples for nanoscopy <https://www.nature.com/articles/nphoton.2009.2>. Both classes of nanoparticles are bright and stable photo-emitters.

In the new version of the paper that entire section has been removed. For the sake of clarification, our statement was correct in the context of super-resolution imaging of quantum emitters – i.e. single photon sources. High resolution optical imaging of individual quantum emitters has only been demonstrated – so far – for nitrogen-vacancy (NV) centres in diamond.

3) Authors should better discuss the advantages and novelty of their low power modality of GDS and compare it with the standard STED imaging ($10^7\text{W}/\text{cm}^2$) RESOLFT, and MINIFLUX modalities

The Reviewer raises a fair point. We have now revised the manuscript with a broader discussion on super resolution imaging modalities, the non-linear photo-physics of hBN emitters and how this can be used to quantifiably improve the resolution of our newly proposed scheme of GSD nanoscopy.

Reviewer #2

1) Orientation given to the manuscript to propose the use of hBN as label for super-resolution is probably premature. First there are large variations in the photoluminescence spectra recorded across different parts of the sample and thus a large variety in the defect natures. Very few of these appear suitable to truly gain from the high resolution "repumping (@532nm)" imaging discussed. Thus to be usable as label, more must be known on the emitting defects and more must be known on how to prepare these. Targeted studies should be made (I assume these would also involve TEM for example).

This is a fair comment the answer to which is indeed non-trivial. As we now emphasize in the first, revised part of the manuscript, hBN is presently the object of intense study by numerous groups. Yet questions like the nature of the host defect(s) and the large distribution of zero phonon line (ZPL) energies are still awaiting for a convincing answer. Understanding the origin and ultimately controlling the properties of hBN emitters is the concurrent objective of our future research and of that of other groups in the field. Yet, this is still likely to be a long-term objective despite a massive rate of progress in both emitter characterisation/understanding and hBN functionalization, which brings us to agreeing with the Referee's comment about using hBN for labeling purposes as 'premature'. We have therefore shifted the emphasis of the manuscript to focus on the newly discovered nonlinear photo-physics of this class of hBN emitters and optical control of the emitters, whilst reducing the attention given to the super-resolution imaging aspect of the paper.

2) Second, it is clear that for some few defects the addition of the low power beam at 532 nm truly helps to reach saturation but this still requires tens of mW in the primary beam which is not really advantageous versus current STED.

The scope of our work is *not* to propose a new super-resolution imaging technique that outperforms current STED methods. Aligning with the Reviewer's and Editor's feedback, we have now emphasized this aspect in the new version of the manuscript. We focus on presenting and discussing the novel nonlinear photo-physics discovered for this class of hBN quantum emitters; we subsequently demonstrate its use for a new modality of far-field, super-resolution nanoscopy. Nevertheless we would like to remark that STED has some intrinsic limitations. In fact in the context of solid-state quantum emitters, only diamond nitrogen-vacancy (NV) centres have so far been imaged beyond the diffraction limit via STED-based methods. Super-resolution imaging of other systems such as diamond silicon-vacancy (SiV) centres, as well as other emitters in silicon carbide and zinc oxide is beyond reach – though possible with other approaches (e.g. stochastic methods based on fluorescence intermittency or 'blinking'). Conversely, the technique we propose exploits optical nonlinearities that might be present (even with different characteristics) in quantum systems other than just this newly reported class of hBN emitters. In this regard, we believe our proposal could lead to alternative – complementary – super-resolution imaging methods for various quantum systems.

3) Third, the resolution is at the end not better than STED. Finally, how would "pieces" of hBN be attached to detection proteins or other biomolecules (without altering the photo-physical properties of the hBN defects involved).

As mentioned in the previous point, our scope is not to propose a technique superior to STED nanoscopy, but rather to advance an alternative technique that exploits the interesting photo-physics we discovered for these hBN emitters – which, as we discussed in (2), could also be extended to other systems.

Having said that, functionalization of hBN has been studied (Weng et al., *Chem. Soc. Rev.*, 2016, 45, 3989). To our knowledge so far, quantum emitters in hBN are extremely stable, and their optical properties are maintained under various environments that include high temperatures (at least 850 K) and very harsh chemical environments [Kianinia M, et al. *ACS Photonics* 4.4, 2017, 768-773 and Tran, Toan Trong, et al. *ACS nano* 10.8, 2016, 7331-7338].

4) It is unknown whether the scheme can be generalised to other materials (ie, adding the low power green to reach saturation with less primary power). It must be pointed out as well, that the literature contains already many works where variations of RESOLFT have been proposed/demonstrated and where the typical STED (or here GSD) beam scheme is modified to

probe various materials including non-fluorescent ones (STED/PALM and derivative methods are already covering very well the fluorescence).

There is no intrinsic limitation to applying the scheme we propose to other systems, provided they possess photo-physical properties analogous to those of the hBN emitters investigated in this study. The properties can be identified using the methodology outlined in our manuscript.

5) I find the photoluminescence analysis to be very well done throughout the paper. It is perhaps needed to indicate the temporal structure of all the lasers used. The list of laser sources is quite extensive and it is unclear which laser is used to record a given datasets. This would be helpful for readers and rather easy to add.

We thank the Reviewer for their commendation on the quality of our analysis. We now clarified the type/wavelengths of the laser sources and where they were used in the “Methods” section.

6) I have no doubt that the resolution limit is “broken” in this manuscript. Yet, it is essentially achieved via the GSD scheme introduced by S. Hell a decade or so ago. The primary doughnut excitation images show this clearly and is responsible for most of the super-resolution. The novelty lies in the addition of the weakly powered 532 nm doughnut. Although the effect is impressive for some of the defects that are probed, the overall gain in terms of resolution is relatively minor at the end, since the system seems to be quite photo-stable anyway (basically the order of magnitude of powers are still the same as for standard GSD). Thus at the end one has a GSD scheme as defined by S. Hell with a defect-dependant power economy. Thus also the idea of achieving super-resolution at lower power is overall desirable, it is unclear if it is achieved on a materials that effectively needs it here. Fig3f shows that the resolution with and without the additional doughnut are essentially converging (when increasing the primary power) experimentally to a same value. It would be interesting for the authors to consider discussing what would happen if the primary power were to be increased further (would the 2 curves effectively exactly match?)

As mentioned above, the focus of the manuscript has now been shifted to emphasize the emitters’ nonlinearity, which the Reviewer acknowledges as important. Moreover, while the ultimate resolution is marginally improving with the addition of the green laser, at lower power the overall emission from the hBN dramatically increases. This is an important attribute that is of a paramount importance for many applications. Also, as seen in Figure 3f for low powers, the addition of only 10 uW of a 532-nm laser improves the resolution by a factor of two. This is highly beneficial for super-resolution, whilst not ‘mandatory’. Nevertheless the simplicity of our method makes it appealing.

Regarding the last point raised by the Reviewer, in the revised manuscript we have added an explanation for the expected behavior of the resolution limits at very high powers. The Reviewer suggests that at high power both the scheme with and without the repumping laser would converge to a common resolution limit. The resolution is ultimately defined by the ratio I/I_s , I being the excitation intensity and I_s the saturation intensity. The repumping laser has the effect of reducing the (absolute) value of the saturation intensity (I_s). As the excitation intensity I increases its value might indeed dominate over I_s : this occurs when $I \gg I_s$ and hence the advantage of having a reduced I_s is limited. Yet, theoretically, the values of the resolution with and without repumping will never be the same (for the same value of I), at any given power, as the repumping laser will always make I_s comparatively smaller.

7) In Fig2d, the saturation curve for 10 microW seems to decrease at higher 708nm power? Why is that? This could be detrimental to the super-resolution should the resolution be pushed further.

The reason for this is unknown. A similar trend has been reported for nitrogen-vacancy (NV) centres in diamond, though the physical origin of the mechanism is still unclear (Han, K. Y., et al. "Dark state photo-physics of nitrogen–vacancy centres in diamond." *New Journal of Physics* 14.12, 2012, 123002). The current limited knowledge on the nature of hBN emitters does not allow concluding on whether or not they display the same behaviour of diamond NV centres.

8) Line 171 and methods. The authors use a “deconvolution”. The methodology described seems to be a high pass spatial frequency filter (ie, the low frequencies are removed when subtracting the smoothed doughnut image). It is unclear that this method would work in general (in fact, the simulations in Fig4d suggest that artefacts will appear already with 2 identical objects). The operation here seems to be a differentiation (thus analogous to S. Hell’s GSD scheme) but with low pass filtered doughnut image instead of a Gaussian-PSF image. The subtraction is not a bad operation here (if one is careful to the potential formation of sidelobes in the subtracted image where the doughnut and reference (usually Gaussian) do not match well at the edges. There are subtraction algorithms that have appeared recently that should handle this issue. As currently used here, it is unclear that the “deconvolution” here would work well on large image involving more than on emitter. It is also not fully clear that the operation eventually lead to a proper deconvolution. In my view, the “deconvolution” used here would have to be seriously “upgraded” for the scheme to work on specimen that involve more than very sparse emitters. How would the method fare for an image such as the one presented in the first figure in the SI?

We used the definition “deconvolution method” based on the homonymous method presented in S. Hell group’s recent work on GSD nanoscopy (Oracz, J., et al. "Ground State Depletion Nanoscopy Resolves Semiconductor Nanowire Barcode Segments at Room Temperature." *Nano Letters* 17.4, 2017, 2652-2659). For clarity, we have added the corresponding reference in the manuscript. The reference contains an extensive discussion of this “deconvolution method” and covers the points raised by the Reviewer. We acknowledge that it is possible to optimize our subtraction algorithm to achieve better results, but we want to emphasize that this is not the scope of our current work – even more so now that the focus of the paper has been shifted towards the photo-physics and nonlinear dynamics of this class of hBN emitters.

9) Line 212-213: “The highest resolution that we measured using the one- and two-laser excitation schemes is (63 ± 4) nm and (87 ± 10) nm, respectively”. This is probably the other way around.

The reviewer is correct. This is now fixed.

10) Fig4: There is very little details on the modelling of the line profile shown and little details about the parameters involved (power? beam fwhms? wavelengths? relaxation times? how close are the emitters? etc), perhaps more could be written into the SI. Moreover when looking at Fig4d, it is not obvious to say that the “repumping” affords a better resolution than without it? (line253-254). Both emitters are quite distinct in all the cases shown (and all dip half-way more or less, in both cases). Maybe the statement line 253-254 and line 270-271 should be clarified and providing more info on the simulations may help with that. Just to be clear, I agree with the

statement line 270-271 but I am not convinced that this is shown in the simulations. Thus the simulations do not appear to be truly supporting the manuscript.

In the revised manuscript, a detailed explanation has been added in the Methods section outlining the details of the simulation conducted in our work.

11) Methods: Sample preparation: the authors probably mean hBN and not graphene?

'Graphene Supermarket' is the name of the company we purchase the hBN flakes from. As confusing (and unfortunate) as it sounds, it is just the name of the supplier.

Reviewer #3

1) The sentence starting with Furthermore in the introductory paragraph is not clear. I now know what they wanted to say after having read the manuscript, but this sentence confuses the reader. The nonlinear photophysical property is perhaps more indicative of a nonlinear optical response rather than a saturation-based effect.

We agree with the Reviewer, the sentence has now been removed.

2) Line 56: Is there a reason why the authors use quantum emitters and SPEs separately in the same sentence? It sounds like they are different entities and photostability of one affects the resolution of the other.

The expressions 'quantum emitters' and 'single photon emitters (SPEs)' are for all intents and purposes synonyms. In the new version of the manuscript, only the term 'quantum emitters' is used to avoid confusion.

3) Line 64: The authors use (in many other parts too) phrases like "very high" or "low" and it is not clear from the manuscript to what these refer. For example are they in the end showing an order of magnitude reduction in the needed laser power, or just a factor 2?

In the revised version of the manuscript we have removed and or specified the expressions the Reviewer refers to. Regarding the reduction in laser power, we now discuss this in more details (First two paragraphs after Equation 1). The resolution is ultimately defined by the ratio I_s/I_m in Equation 1. We clarify the improvement obtained with our method using a numerical example (Third paragraph after Equation 1) which shows that an addition of only 10 uW of the 532-nm laser (doughnut-shaped) produces a twofold improvement in resolution compared to the single-doughnut excitation, but this can be further increased by optimizing the combination of powers as per Equation 1.

4) Figure 1 panel c: The authors do not mention the solid curve (fit) to the data. Is there a background in the intensity? It might be good to mention the fitting function.

The data has been fitted to the saturation equation routinely used in fluorescence microscopy ($I = I_\infty \times P / (P + P_{\text{sat}})$). We have included the explanation in the caption of the figure as well as in the main text of the revised manuscript.

5) Figure 1 panel d: Why do the authors pick 10 uW and 300 uW for the two lasers? Are these special values (i.e. saturation etc)

The main aim of Figure 1d is to highlight the nonlinear optical behaviour of the emitters, which occurs under co-excitation with the laser pair (675-nm and 532-nm). The powers were thus

arbitrarily – yet purposefully – chosen to emphasize that a linear increase in excitation power corresponds to an enhanced, nonlinear photoluminescence signal from the emitter. Specifically, going from [300 uW_{@675-nm}]-excitation to [300 uW_{@675nm} + 10 uW_{@532-nm}]-excitation (i.e. an increase in excitation of 3.3% assuming, in first approximation, that the absorption is wavelength-independent) produces an increase in fluorescence by a factor above two.

This is also explained in the main text, appearing just before Figure 1.

6) Figure 1 panel e: I have a hard time seeing what the authors want me to see here. The image gets brighter by laser power, but the nonlinearity is far from obvious. I suggest an intensity plot like panel c and an inset image of the spatial resolution as an example.

Figure 1 has been modified according to the Reviewer's suggestion.

7) Figure 2 panels a and b: Isn't it possible to extract from the autocorrelation measurements a power-dependent τ_1 to compare with panel b? The dynamics should reveal the τ_1 one expects and contrast it to the measured data.

It is indeed possible. Panel b shows the extracted τ_1 and τ_2 from the autocorrelation data. Judging from the Referee's comments, it appears that the labeling on the figure might have been misleading. The labeling as well as the caption of the image has been revised according to the Reviewer's comment.

8) Finally, from my perspective, the manuscript offers a new technique for superresolution using a class of emitters matching the internal level scheme of the hBN quantum emitters. This is the bigger fish, the fact that we can locate the quantum emitters with better resolution is less important. The main reason for this is that imaging these quantum emitters with 60-80 nm resolution as a main result has less appeal because this is not a particularly attractive length scale to either reveal the internal structure of the quantum emitters or have superior localisation in a photonic chip. Simple high-quality confocal imaging and inference allows for ~20-40 nm localisation already, so the authors should at least beat that if they want this to be their main result. My main suggestion would be a revision of the manuscript to put the new technique more in the spotlight.

As mentioned at the beginning of this Response Letter, we find this comment to be particularly relevant. As a result, we have restructured our manuscript shifting its emphasis on the newly discovered, nonlinear photo-physics of this class of hBN emitters – followed by its consequent exploitation for a new modality of super-resolution imaging.

Reviewer #1 (Remarks to the Author):

I would like to thank authors for submitting a revised version of the paper. Although the overall quality of the paper improved, including data related to the photo-physics of the single emitter, authors failed to demonstrate resolution improvement in full by resolving closely spaced emitters. I believe that reason for this originates from the material used in this work.

The authors discuss two-dimensional materials and layered materials at many places in the paper including the title. However, none of the relevant results in this work is found to be layer dependent or with confidence observed in a single layer. For example, is the interesting photophysics dependent on hBN layer numbers? More specifically,

1. First, the authors used a coating of small flakes to the substrate. In the referred publication ACS Photonics 4.4, 2017, 768-773 from the authors 'group (which used same methods for preparing samples), the height profile of only 3 selected 'flakes' was shown and statistics of the averaged thickness of hBN samples has not been systematically discussed. These 'flakes' might aggregate during the coating procedure.
2. The size of the hBN flakes. Referring to the only AFM image shown in the supporting information of ACS Photonics 4.4, 2017, 768-773, hBN flakes are typically smaller than 100 nm and some are even smaller than 10 nm. Reduction of the lateral size of layered BN sheets could also form BN quantum dots. In this case, not only defect centers plays a key role, but also zigzag edges structures could support the observed photoluminescence, as found in graphene quantum dots.
3. Samples prepared by mechanical exfoliation of hBN crystals or CVD (typical size in um range) can rule out these possibilities. In order to support their claims, I suggest the author to use mechanically exfoliated or CVD sample (which is also available from graphene supermarket <https://graphene-supermarket.com/Singlelayer-h-BN-Boron-Nitride-film-grown-in-copper-foil-2-x-1.html>) to perform same measurements.
4. In addition, the color diversity of defects in h-BN ((19) Tran, et al. ACS Nano 2016, 10, 7331–7338. Jungwirth, N. R.; et al. Nano Lett. 2016, DOI: [10.1021/acs.nanolett.6b01987](https://doi.org/10.1021/acs.nanolett.6b01987) will render the used of STED imaging impractical since we lack control on the position, type and consequently spectral response of the emitter

The demonstration of STED imaging is still not complete, the authors managed to use depletion beam to shrink the confocal spot to 70 nm but no super-resolution imaging has been achieved in this work as pointed by other reviewers. Localization of the emitter center is not the same as super-resolution imaging. One would still need to construct the image by scanning samples in the case of STED and sequential imaging for STORM/PALM. An

example has been given in the Nature Photonics paper from Stefan Hell. The current work still fails to achieve STED imaging as shown in Figure 2b.

Figure 2 | Stimulated emission depletion microscopy reveals densely packed nitrogen-vacancy centres in diamond. **a,b**, Confocal (**a**) and STED (**b**) images from the same crystal region. **c**, The individual centres resolved in **b** automatically yield the effective PSF of the STED recording whose y -profile exhibits a FWHM of $\Delta y = 16.0$ nm. **d**, The coordinate of each centre can be calculated with 0.14 nm precision. Comparing **a** with **d** highlights the dramatic gain in information resulting from the unique increase in resolution. **e,f**, Applying $I_{\text{STED}}^{\text{max}} = 3.7 \text{ GW cm}^{-2}$ shrinks a confocal spot of 223 nm diameter (FWHM) down to 8 nm. Note that the increase in resolution is a purely physical phenomenon.

(Figure from Rittweger, E., Han, K. Y., Irvine, S. E., Eggeling, C., & Hell, S. W. (2009). STED microscopy reveals crystal color centers with nanometric resolution. Nature Photonics, 3(3), 144-147.)

Reviewer #2 (Remarks to the Author):

Having read the authors' rebuttal and their revised manuscript, I am pleased to find that the authors have strongly shifted the emphasis of their manuscript towards the peculiar photo-physics found in some hBN defects and kept the super-resolution application as one direct application/illustration of their findings. In this case, the shortcomings arising from the lack of robust super-resolution image recovery algorithm, from the limited super-resolution wrt STED, GSD and other methods (including localization) are less relevant indeed.

Overall, I find that the observations raised earlier and have been well addressed by the authors.

The model used for Fig. 4 is now properly described in the manuscript. Yet, line 301-302, I would still downplay the comparison between model and experiment. For me the calculation cannot "match well" with the data, because the parameters used are in fact chosen (now described in the methods) so that this is indeed the case. The "match" is quite guided and not independent from the experiment. For me, the model illustrates that the relationship between super-resolution (enhanced by the repumping) makes sense qualitatively but nothing beyond. In fact this is actually what the authors states in the method section where they now describe the calculation. Mind that the experimental data are on the other hand sufficiently convincing that the qualitative match found with the model is plenty enough for this paper.

I find this revised paper well done. A drawback is on the other hand the lack of knowledge and control on the defects; but the authors have already indicated in their rebuttal that this would require a longer effort from them as well as from community. A pity because this would really push the paper much further (essentially, from the paper, we know that this nice photo-physics can happen randomly and sparsely in a hBN sample. But we just do not know enough about the defect structure(s) involved for a thorough understanding (and thus generalization to other materials) and reliable preparation/exploitation in hBN.

Perhaps also missing at the end of the paper is an introduction/discussion to other potential applications of the property found on hBN, beyond super-resolution labels.

Reviewer #3 (Remarks to the Author):

I am happy with the revisions the authors have done. The manuscript is more coherent and the claims are more appropriate. I recommend it for publication.

Reviewer #2

Having read the authors' rebuttal and their revised manuscript, I am pleased to find that the authors have strongly shifted the emphasis of their manuscript towards the peculiar photo-physics found in some hBN defects. Overall, I find that the observations raised earlier and have been well addressed by the authors.

We thank the reviewer for the positive assessment.

The model used for Fig. 4 is now properly described in the manuscript. Yet, line 301-302, I would still downplay the comparison between model and experiment. For me the calculation cannot "match well" with the data, because the parameters used are in fact chosen (now described in the methods) so that this is indeed the case. The "match" is quite guided and not independent from the experiment. For me, the model illustrates that the relationship between super-resolution (enhanced by the repumping) makes sense qualitatively but nothing beyond. In fact, this is actually what the authors states in the method section where they now describe the calculation. Mind that the experimental data are on the other hand sufficiently convincing that the qualitative match found with the model is plenty enough for this paper.

We agree with the Reviewer and modified the last sentence before Figure 4 (line 302/3) to: *"Our calculations provide a qualitative match to the experimental results (figure 3d – f)."* In addition, we have indicated the transitions in figure 2c for more clarity.

I find this revised paper well done. A drawback is on the other hand the lack of knowledge and control on the defects; but the authors have already indicated in their rebuttal that this would require a longer effort from them as well as from community. A pity because this would really push the paper much further (essentially, from the paper, we know that this nice photo-physics can happen randomly and sparsely in a hBN sample. But we just do not know enough about the defect structure(s) involved for a thorough understanding (and thus generalization to other materials) and reliable preparation/exploitation in hBN.

We acknowledge this point and note that the amount of interest in hBN and rate of progress in understanding the defects is extremely high. It therefore seems likely that all of these issues will be resolved in due course - as has been the case for other materials such as diamond, that have been studied intensely by a diverse international research community. We trust our results will expedite this research by stimulating more groups to join this effort.

Perhaps also missing at the end of the paper is an introduction/discussion to other potential applications of the property found on hBN, beyond super-resolution labels.

This is a valid point. Indeed, our scheme can be widely used to improve brightness of emitters, but more importantly, to minimize spectral diffusion. The former is important to realize resonant excitation and indistinguishable photons, which is crucial for employing these defects in quantum information. We added the following line to the conclusion: *"The nonlinear behavior and the repumping mechanism can also be used to suppress spectral diffusion and thus aid with the generation of indistinguishable photons from single photon emitters in hBN"*.

Reviewer #1

I would like to thank authors for submitting a revised version of the paper. Although the overall quality of the paper improved, including data related to the photo-physics of the single emitter, authors failed to demonstrate resolution improvement in full by resolving closely spaced emitters. I believe that reason for this originates from the material used in this work.

The authors discuss two-dimensional materials and layered materials at many places in the paper including the title. However, none of the relevant results in this work is found to be layer dependent or with confidence observed in a single layer. For example, is the interesting photophysics dependent on hBN layer numbers?

Our aim was not to resolve two closely spaced emitters, neither was it to study the effect of layer number on emission. This is an entirely separate (though interesting) research direction. Our goal, as stated clearly in the manuscript, was to study the new non-linear effect in hBN and exploit this effect in the design of a new modality of GSD. Our use of the word "layered" is valid - hBN is a layered van der Waals material.

More specifically,

1. First, the authors used a coating of small flakes to the substrate. In the referred publication ACS Photonics 4.4, 2017, 768-773 from the authors 'group (which used same methods for preparing samples), the height profile of only 3 selected 'flakes' was shown and statistics of the averaged thickness of hBN samples has not been systematically discussed. These 'flakes' might aggregate during the coating procedure.

As mentioned earlier the geometrical effects are not considered or studied in our work. The aggregation aspect is also not important, as we probe only a single emitter as evidenced by our second order autocorrelation measurement. (fig 1b inset)

2. The size of the hBN flakes. Referring to the only AFM image shown in the supporting information of ACS Photonics 4.4, 2017, 768-773, hBN flakes are typically smaller than 100 nm and some are even smaller than 10 nm. Reduction of the lateral size of layered BN sheets could also form BN quantum dots. In this case, not only defect centers plays a key role, but also zigzag edges structures could support the observed photoluminescence, as found in graphene quantum dots.

The reviewer is confusing the nature of the studied emitters with quantum dots. The emitters are deep defects (not ionized even at 800 K, as shown in the above-cited ACS Photonics paper) in the 6-eV bandgap of hBN. There is no evidence of quantum confinement effects in any of the hBN literature on the defects, and the valence/conduction bands are not involved in the light emission process.

3. Samples prepared by mechanical exfoliation of hBN crystals or CVD (typical size in um range) can rule out these possibilities. In order to support their claims, I suggest the author to use mechanically exfoliated or CVD sample (which is also available from graphene supermarket) to perform same measurements.

Again, this comment is not relevant, as per our above response.

4. In addition, the color diversity of defects in h-BN ((19) Tran, et al. ACS Nano 2016, 10, 7331–7338. Jungwirth, N. R.; et al. Nano Lett. 2016, DOI: 10.1021/acs.nanolett.6b01987 will render the used of STED imaging impractical since we lack control on the position, type and consequently spectral response of the emitter.

Our work is not on STED, but instead GSD. But regardless, this comment from the referee is pure speculation. There is no reason to believe that the color-diversity of emitters in hBN will not be understood in due course. Similarly, there is no reason to believe that current efforts aimed at deterministic engineering of monochromatic emitters in hBN will fail. Lastly, if and

when these efforts succeed, the color-diversity will in fact be desirable for applications of super-resolution imaging based on emitters in hBN (e.g. multiplexing).

The demonstration of STED imaging is still not complete. One would still need to construct the image by scanning samples in the case of STED and sequential imaging for STORM/PALM. An example has been given in the Nature Photonics paper from Stefan Hell. The current work still fails to achieve STED imaging.

The focus of the work is on understanding a new non-linear effect, and exploitation of this effect in super-resolution imaging. Our demonstration of resolution beyond the diffraction limit is technically sound, as also remarked by Referee #2: "There is no doubt that the diffraction limit has been broken". We emphasize, again, that our work is on GSD, not STED (see point 4).